# Cultivation Conditions of Spinach and Rocket Influence Epiphytic Growth of *Listeria monocytogenes*

**DOI:** 10.3390/foods11193056

**Published:** 2022-10-01

**Authors:** Paul Culliney, Achim Schmalenberger

**Affiliations:** Department of Biological Sciences, University of Limerick, V94 T9PX Limerick, Ireland

**Keywords:** *Listeria monocytogenes*, leafy vegetables, variety, polytunnel, open fields, seasonality

## Abstract

Leafy vegetables are associated with Listeriosis outbreaks due to contamination with *Listeria monocytogenes*. To date, contradictory findings were reported on spinach, rocket, and kale, where some studies reported growth of *L. monocytogenes*, while others did not. Thus, the current study investigated the reason for conflicting findings by producing leafy vegetables, where cultivation factors were known for growth potential studies. Of all polytunnel produce, kale Nero di Toscana demonstrated the highest growth potential (2.56 log cfu g^−1^), followed by spinach F1 Cello (1.84 log cfu g^−1^), rocket Buzz (1.41 log cfu g^−1^), spinach F1 Trumpet (1.37 log cfu g^−1^), and finally rocket Esmee (1.23 log cfu g^−1^). Thus, plant species and variety influenced *L. monocytogenes* growth potentials. Moreover, significantly lower growth potentials of 0.3 log cfu g^−1^ were identified when rocket Buzz was cultivated in open fields (1.11 log cfu g^−1^) instead of a polytunnel. The opposite effect was observed for spinach F1 Trumpet, where growth potentials increased significantly by 0.84 log cfu g^−1^ when cultivated in open fields (2.21 log cfu g^−1^). Furthermore, a significant seasonality effect between batches was found (*p* < 0.05). This study revealed that spinach and rocket cultivation conditions are at least co-factors in the reporting of differing growth potentials of *L. monocytogenes* across literature and should be considered when conducting future growth potential studies.

## 1. Introduction

As of February 2022, the Centers for Disease Control and Prevention (CDC) is investigating two separate recalls (December 2021 and January 2022) of packaged leafy vegetables due to *L. monocytogenes* contamination [1]. Thus far, one ongoing *L. monocytogenes* outbreak has been linked with 13 hospitalisations and two deaths. The consumption of ready-to-eat (RTE) leafy vegetables has been linked with further recent *L. monocytogenes* outbreaks [2,3]. Consumer demand for RTE leafy vegetables with variations in organoleptic properties including taste, shape, and colours have led to the development and increasing popularity of alternative varieties of baby leafed RTE vegetables [4]. In Mediterranean regions in the late autumn–winter season, plants grown in open-fields present more desirable traits for rocket with increased desirable colour and better shape compared to greenhouse produce. In cooler regions with intense rain, leafy produce in open fields are negatively impacted by producing undesirable traits [5]. In countries where leafy vegetables are not available due to challenging weather conditions or during the winter season, polytunnels are utilised to meet the demand for fresh leafy vegetables. Therefore, both cultivation methods are used interchangeably depending on the farm, country, and season.

Smith and colleagues (2018) provided an example of the fresh leafy produce supply chain produced in an open field setting and revealed several potential contamination routes for *L. monocytogenes* [6]. Due to the ubiquitous nature of *L. monocytogenes*, contamination can occur anywhere throughout the fresh produce supply chain [7,8]. Rain and irrigation events, manual handling, proximity to roads and urban developments, presence of wildlife and livestock manure (faeces), i.e., organic fertilisers, have been linked with increased incidences of *L. monocytogenes* contamination [3,9,10,11,12]. A 17.5% prevalence rate of *L. monocytogenes* was established on fresh produce fields in New York State [11]. The same authors revealed that manure application one year prior to sample collection was associated with seven times greater odds of *L. monocytogenes* isolation. *L. monocytogenes* presence was confirmed on 14 fresh leafy vegetable samples (leaf lettuce, head lettuce, mixed greens, spinach, etc.) out of 4435 sold at retail level from 2009 to 2013 in Canada (prevalence rate of 0.32 %) [13]. 

*L. monocytogenes* has the ability to survive and grow within harsh food preservation conditions such as temperatures from −1.5 °C to 45.0 °C, salt concentration of up to 16%, water activity levels as low as 0.93 and pH levels from 4.2 to 9.5 [14]. The influence of environmental conditions on *L. monocytogenes* on leafy vegetables has been well-established. The application of different atmosphere treatments has been indifferent to the limited impact of the growth of *L. monocytogenes* on leafy vegetable products [15,16,17]. Incubation over 6 days at 7 °C, combination of 7 °C and 15 °C, and solely 15 °C showed increasing growth potentials of *L. monocytogenes* on spinach and rocket with an increase in temperature [18]. The same study reported that *L. monocytogenes* growth on leafy vegetables could be controlled during commercialisation by maintaining temperatures under 7 °C. *L. monocytogenes* and growth potentials were higher at 8 °C compared to 5 °C with rocket supporting growth (>0.50 logs) at 8 °C and iceberg lettuce at both temperatures [19]. Likewise, baby spinach displayed considerably higher growth at 9 °C compared to 4 °C [20]. An additional lower initial inoculation density of 1 log *L. monocytogenes* was associated with greater growth potentials on lettuce, spinach, and rocket, compared to 2 logs [21]. Moreover, the same study revealed that there are factors other than incubation conditions that influence the development of *L. monocytogenes* on RTE leafy vegetables. However, cultivation conditions were not revealed on the packaging of the vegetables purchased from the supermarket. Moreover, contradicting data of *L. monocytogenes* on spinach and rocket still exists with some supporting growth [21], while others did not [22,23]. Additional studies identified conflicting trends of *L. monocytogenes* growth with rocket supporting more growth than spinach [18], whereas a recent study found more growth on spinach compared to rocket [21]. Kale had no significant changes over 6 days in *L. monocytogenes* populations when inoculated with 4 logs at 4 and 13 °C [24]. Another study with an initial inoculation density of 5 logs at 7 °C led to a relative increase of 1.2 to 1.5 logs *L. monocytogenes* growth on kale after 10 days [22]. Consequently, there is a need to determine if kale should be categorised as being able to support *L. monocytogenes* growth by conducting challenge studies that follow EU guidelines for experimental conditions, i.e., 2 log *L. monocytogenes* inoculation density (three-strain mix), combined with a 7, 12 °C temperature profile. 

According to EU regulation 2073/2005, for RTE foods that support growth of *L. monocytogenes*, food business operators (FBOs) must demonstrate the absence of the food-borne pathogen in five samples of 25 g. However, for RTE foods that do not support growth, up to 100 cfu g^−1^
*L. monocytogenes* is allowed throughout the shelf-life of that product. If the inability to support *L. monocytogenes* growth has not been demonstrated, then growth is presumed. Therefore, EU reference laboratory’s guidance document outlines experimental guidelines which must be followed by accredited laboratories on behalf of FBOs to determine whether their food products can support the growth of *L. monocytogenes* [14]. The same document provides guidelines for calculating the maximum growth rate, which can be exploited to calculate the growth rate at another temperature on the same food product [25].

The aim of the current study was to define the influence of leafy vegetable growth conditions on *L. monocytogenes* growth. At outset, our hypothesis was that cultivation conditions influence the growth of *L. monocytogenes* and, consequently, are responsible for the conflicting growth reported on rocket and spinach. Thus, in efforts to resolve this inconsistency, cultivation factors, i.e., vegetable species and variety, open field vs. polytunnel cultivation, and seasonality, were tested on the growth of *L. monocytogenes* on leafy vegetables. 

## 2. Materials and Methods

### 2.1. Production of Leafy Vegetables (Experimental Farm Produce)

Rocket (Buzz), rocket (Esmee), spinach (F1 Trumpet), spinach (F1 Cello), and kale (Nero di Toscana) seeds were sown one inch deep in John Ines No.1 compost in covered modular trays. The trays were then placed in a polytunnel for 2 weeks prior to transplantation in either a polytunnel or open field setting until harvest. Produce were harvested in their final stage of leaf development. The time from sowing the seeds to harvest ranged from 35 to 56 days, with an average of 40.8 days (Appendix A). Growing periods were as follows: polytunnel rocket (Buzz) batch 1 = 17th June to 22nd July, batch 2 = 1st July to 5th August, and batch 3 = 15th July to 19th August; open field rocket (Buzz) batch 1 = 10th June to 22nd July, batch 2 = 24th June to 5th August, and batch 3 = 8th July to 19th August; polytunnel spinach (F1 Trumpet) batch 1 = 17th June to 22nd July, batch 2 = 1st July to 5th August, and batch 3 = 15th July to 19th August; open field spinach (F1 Trumpet) batch 1 = 24th June to 5th August, batch 2 = 8th July to 19th August, and batch 3 = 11th September to 4th November; polytunnel rocket (Esmee) 5th March to 20th April; polytunnel spinach (F1 Cello) 5th March to 20th April; and Kale (Nero di Toscana) 28th April to 9th June. Weather data during the growing periods were recorded from the local weather station; for outdoor and for indoor production, a temperature logger was used to record maximum and minimum temperatures within the polytunnel. Average weather data were recorded, i.e., maximum air temperature (°C), minimum air temperature (°C), precipitation amount (mm), and sunshine duration (hours) (Appendix A). 

### 2.2. Preparation of L. monocytogenes for Inoculation of Leafy Vegetables

For growth potential experiments, three different strains of *L. monocytogenes* from the Teagasc Food Research Centre strain collection (Moorepark, Ireland) were used: 959 (vegetable isolate), 1382 (EUR Lm reference strain), and 6179 (food processing plant isolate). For each of the three *L. monocytogenes* strains, 10 mL of autoclaved tryptone soya broth (TSB, CM0129, Oxoid, Basingstoke, UK) was prepared and placed in sterile 50 mL conical flasks. Single colonies from the previously streaked plates (*Listeria* selective agar, conforming to ALOA, Chromocult^®^ LSA, Merck, Darmstadt, Germany) of *L. monocytogenes* culture were transferred into each flask and incubated at 7 °C for 7 days. Spectrophotometry was used to verify the cell density (600 nm) (Biochrom, Cambridge, UK). Dilutions with phosphate-buffered saline (PBS, pH 7.3, BR0014, Oxoid) were carried out to mix the three strains at equal cell densities to aim for inoculation at cell densities of 100 cfu g^−1^. This was confirmed by enumeration on *Listeria* selective agar (LSA), conforming to ALOA [17,21,26].

For maximum growth rate experiments, single strain inoculums were required for inoculation of the produce. EURL guidelines required two *L. monocytogenes* strains to be tested separately with initial inoculum of 100 cfu g^−1^ for each product [14]. *L. monocytogenes* strains 959 (vegetable isolate) and 1382 (EUR Lm reference strain) were prepared identically as per growth potential experiments reported above. However, strains were not combined. 

### 2.3. Preparation of the Polypropylene Bags

Oriented polypropylene packaging film (35 μm thick) was used to create storage bags (18 cm × 10 cm) with a permeability to O_2_ of 5.7 nmol m^−2^ s^−1^ kPa^−1^ and to CO_2_ of 19 nmol m^−2^ s^−1^ kPa^−1^ (Amcor Flexibles, 120 Gloucester, UK). For the growth potential experiments, 68 bags were required for each product batch to allow for testing in quadruplicates of *L. monocytogenes* and total bacteria count sampling on days 0, 2, 5, 7, and 9 (20 packages); total bacteria counts (TBCs) of non-inoculated test units (20 packages); pH and water activity measurements of non-inoculated test units (20 packages); and absence of *L. monocytogenes* on day 0 and day 9 in control samples (8 packages). For the maximum growth rate experiments, a total of 74 bags (37 per strain) were required for each product batch to allow for testing of *L. monocytogenes* (13 packages per strain per product), TBCs of non-inoculated test units (8 packages per strain per product), pH and water activity measurements of non-inoculated test units (8 packages per strain per product), and for control samples (8 packages per strain per product).

### 2.4. Preparation and Subsequent Inoculation and Storage of Leafy Vegetables

On the day of each experiment, within 24 h of the conducting growth potential experiments, experimental farm produce were harvested, i.e., rocket (Buzz and Esmee), spinach (F1 Trumpet and F1 Cello), and kale (Nero di Toscana). No further produce processing was carried out (e.g., shredding, washing, or chlorine dipping). Twenty-five grams of each product were placed into the prepared polypropene packages (Section 2.3). Using the previously prepared *L. monocytogenes* dilutions (Section 2.2), 100 µL of *L. monocytogenes* three-strain cocktail suspension (representing 100 cfu g^−1^ of food product) was distributed uniformly over the 25 g of leafy vegetable product within the polypropene bags (eight control bags were treated with 100 µL sterile PBS) [21]. The packages were sealed in ambient gaseous conditions using a vacuum packer (Multivac, Dublin, Ireland). For growth potential studies, experiments were conducted at 7 °C ± 0.5 °C (6 days), followed by 12 °C ± 0.5 °C (3 days) (HR410, Foster Refrigerator, King’s Lynn, UK) [14].

On the day of each maximum growth rate experiment, polytunnel spinach (F1 Trumpet) and rocket (Buzz) were harvested and stored under refrigeration conditions until use. The same amount of produce as in the growth potential experiments was used (25 g). Previously prepared single *L. monocytogenes* dilutions (strains 959 and 1382, as described in Section 2.2) were used for maximum growth rate experiments and were inoculated onto produce in the same manner as in the growth potential experiments. The same ambient atmospheric conditions were applied in the polypropylene bags (see Section 2.3). The maximum growth rate experiments were conducted at constant temperature of 8 °C ± 0.5 °C (HR410) for 10 days [14]. 

### 2.5. Product pH and Water Activity and Subsequent Batch Determination

Product pH and water activity were determined in quadruplicates on days 0, 2, 5, 7, and 9 for each product, and average values and standard deviations were reported. For pH measurements on homogenates of each product, a calibrated pH probe (Cole-Parmer, Saint Neots, UK) was used following ISO 1842:1991 procedure for determination of pH of fruits and vegetables [27]. Twenty-five grams of produce were homogenised in a stomacher (Seward 400, AGB Scientific, Dublin, Ireland) for 120 s at a high speed (260 rpm) with an equal mass of distilled water. For determination of water activity values, AQUALAB model Series 3TE water activity meter (LabCell Ltd., Four Marks, UK) was used following the manufacturer’s instructions. Water activity and pH values were input into the inter-batch physico-chemical variability calculator located on EURL’s website to determine their effect on *L. monocytogenes* growth in the tested temperature conditions [14]. 

### 2.6. Sampling of the Leafy Vegetable Packs and L. monocytogenes Analysis

The specific sampling data points for growth potential experiments were days 0, 2, 5, 7, and 9. On each of these days growth of *L. monocytogenes* was determined with four biological replicates with three technical replicates per sample (12 LSA plates). Furthermore, four control bags (without *L. monocytogenes* inoculation) were harvested each on day 0 and day 9, and the absence of *L. monocytogenes* was confirmed on LSA (conforming to ALOA) in accordance with ISO 11290-1 horizontal method for the detection of *L. monocytogenes* and *Listeria* spp. [28]. 

On day 0 and day 9, prior to opening the packs, the concentrations of oxygen were determined inside the bags, using a gas analyser (PBI-Dansensor, PBI Development, Ringsted, Denmark, Model TIA-III LV) with an injection needle to penetrate the packs. Each bag was cut using disinfected utensils (70% iso-propanol), one at a time, directly underneath the heat seal for subsequent sample analysis. The contents of each package were transferred into separate stomacher bags and homogenised using a stomacher (Seward 400), for 120 s at a high speed (260 rpm), in 25 mL of PBS. Depending on anticipated cell counts, samples were concentrated five-fold (via centrifugation at 4000× *g* for 240 s and resuspending in 200 µL PBS; detection limit of 2 cfu g^−1^) or diluted in order to have no more than 200 or less than 10 colonies per plate [21]. Aliquots of 100 µL were then plated on LSA (ALOA) and incubated at 37 °C for 24–48 h. Colony forming units (cfu) on days 0, 2, 5, 7, and 9 were transformed into log_10_ cfu g^−1^, mean values and standard deviations were determined and plotted, and median values were used to calculate growth potentials [14].

Maximum growth rates (μmax) were calculated for spinach (polytunnel; F1 Trumpet variety) and rocket (polytunnel; Buzz variety) and conducted in accordance with EURL guidelines [14]. The sampling data points for determination of maximum growth rate were 0, 24, 48, 72, 96, 120, 144, 168, 192, 216, and 240 h for each curve. At 0 h, per strain per product, three separate test units (biological replicates) were analysed each with nine technical replicates (27 LSA plates). Subsequently, for the remainder of the experiments, at each data point, one test unit was tested with nine technical replicates (nine LSA plates). In addition to EURL guidelines, maximum growth rates were also determined for growth potential curves. ComBase was used to predict the maximum growth rate (Vmax) of *L. monocytogenes* by plotting and fitting *L. monocytogenes* data to linear, biphasic, and Baranyi and Roberts models [29]. Vmax (Log_10_ cfu g^−1^ h^−1^) was then multiplied by 2.3 to obtain the correspondent µmax (Ln cfu g^−1^ h^−1^). R^2^ (also referred to as coefficient of determination) was measured as the goodness of the fit (value of 1 = best fit [30,31]). Root Mean Square Error (RMSE) was the difference between observed and the predicted data with an RMSE close to 0 as the ideal value (predicted and observed data are the same) [32]. 

For *Listeria* sp. colony confirmation, at least one presumptive colony per plate was tested for phosphoribosyl pyrophosphate synthetase (*prs*) gene presence. The presumptive colonies were isolated using sterile toothpicks and lysed using a quick lysis protocol as described [33] and used as a template in the PCR. For PCR, each 25 μL reaction contained 1 × buffer (2 mM MgCl_2_), 0.2 mM dNTP mix, 0.4 mmol of both forward (5′GCTGAAGAGATTGCGAAAGAAG′3) and reverse (5′CAAAGAAACCTTGGATTTGCGG′3) *prs* primers [34], 0.5 U of DreamTaq polymerase (Fisher Scientific, Waltham, MA), and 0.5 μL of template DNA. PCR was performed with an initial denaturation step at 95 °C for 4 min, followed by 35 cycles of 94 °C for 40 s, 53 °C for 30 s, 72 °C for 60 s, and one final cycle of 72 °C for 10 min. PCR products were visualised via electrophoresis as previously described [35]. In the present study, all tested presumptive *L. monocytogenes* colonies (growth on LSA based on ALOA with blue pigment and halo formation) were independently confirmed as *Listeria* via a specific amplicon in the colony PCR.

### 2.7. Total Bacteria Counts (TBCs)

For the growth potential experiments, TBCs were analysed in quadruplicates (four biological replicates) with three technical replicates on days 0, 2, 5, 7, and 9 [21,36]. At each data point, TBCs were conducted on both *L. monocytogenes* inoculated and non-inoculated test units. However, for maximum growth rate experiments per strain and per product, TBCs were determined on day 0 and at day end only. Package contents were transferred into separate stomacher bags and homogenised as described above in 25 mL of PBS. Following this, a dilution series was aseptically carried out with PBS and plated on tryptone soy agar (TSA, CM0131, Oxoid). TBCs were enumerated after incubation at 37 °C for 48 h. ComBase was used to predict the parameters of TBC growth curves (Vmax, R^2^ and RMSE) in each growth potential experiment and µmax, as determined in Section 2.6. 

### 2.8. Statistical Analysis

*L. monocytogenes* and TBCs were reported as the means of four replicates and (±) standard deviations. R-Studio software (version 4.1.1) was used for statistical analysis. For growth potentials of open field versus polytunnel spinach (F1 Trumpet) and rocket (Buzz), growth potentials of spinach (F1 Trumpet) versus rocket (Buzz) (polytunnel and open fields) in situations of normality (Shapiro–Wilk) and homoscedasticity (Levene’s), an independent two-sample equal variance, two-tailed *t*-test was conducted (*t*-tests). Similarly, after the same checks, a one-way ANOVA was conducted to compare the three batches of polytunnel and open field spinach (F1 Trumpet) and rocket (Buzz) for each weather parameter. A normality (Shapiro–Wilk) test was conducted prior to repeated measures analysis of variance (ANOVA) on pH; water activity values of all produce; and *L. monocytogenes* populations and TBCs of rocket (Esmee), spinach (F1 Cello), and kale (Nero di Toscana). Normality (Shaprio-Wilk), homogeneity of covariances (Box’s M), and sphericity (Mauchly’s) tests were performed prior to conducting a mixed model (i.e., between and within design) ANOVA analysis with Tukey post hoc tests. These were carried out on *L. monocytogenes* and TBCs of the remaining produce separately (i.e., polytunnel spinach F1 Trumpet, open field spinach F1 Trumpet, polytunnel rocket Buzz, and open field rocket Buzz) to determine the effect, within factor (time, i.e., repeated measures), and between factor of batches (i.e., seasonality) for *L. monocytogenes* populations and TBCs. Mixed ANOVA was then conducted on all batches in this study separately for plant species (between factor) and time, i.e., repeated measures (within factor) effect on *L. monocytogenes* and TBC populations of spinach (F1 Cello), rocket (Esmee), and kale (Nero di Toscana). 

## 3. Results

### 3.1. Water Activity and pH

Average pH values of spinach variety F1 Trumpet ranged from 6.86 to 7.21, while average water activity ranged from 0.970 to 0.985 between day 0 and day 9. Similarly, for spinach variety F1 Cello, values ranged from 6.82 to 6.99 and from 0.973 to 0.982, respectively. Average pH values of rocket variety Buzz and Esmee ranged from 6.54 to 6.78 and from 6.48 to 6.67, respectively, thus being lower than the pH values from spinach. However, water activities for both rocket varieties were similar to the ones from spinach, from 0.975 to 0.988 for Buzz and from 0.979 to 0.987 for Esmee. Kale (Nero di Toscana) pH values ranged from 6.67 to 6.79, and water activities were between 0.982 to 0.989 throughout the duration of the experiments (Appendix A). The EURL inter-batch variability calculator revealed that the variability of pH and water activity on day 0 for all produce in this study was not significant regarding the growth of *L. monocytogenes* in the tested temperature conditions. Therefore, only one growth potential batch per product was required. Nevertheless, three batches per product were used for *L. monocytogenes* growth potential experiments spinach F1 Trumpet and rocket Buzz to identify potential seasonal effects. Water activities of all produce did not significantly change throughout the course of their growth potential studies and remained close to water activities required for optimal *L. monocytogenes* growth (i.e., 0.990). Moreover, pH of both rocket varieties and spinach F1 Cello significantly changed over the duration of the growth potential studies. This effect was not observed for spinach variety F1 Trumpet or kale. A change in pH values did not affect *L. monocytogenes* growth as the values throughout were close to the optimal growth conditions (i.e., pH 7). 

### 3.2. Influence of Leafy Vegetable Products, Variety, Cultivation Method, and Seasonality of Cultivation) on the Growth of L. monocytogenes on Experimental Farm Produce

All produce used in this study demonstrated the ability to support the growth of *L. monocytogenes* (log_10_ cfu g^−1^ > 0.50) (Figure 1 and Figure 2). A pairwise comparison identified significantly higher growth potentials of polytunnel rocket (Buzz, 1.35, 1.42 and 1.45 log_10_ cfu g^−1^, Figure 1A) over open field rocket (Buzz, 0.97, 1.07 and 1.28 log_10_ cfu g^−1^, *p* = 0.035, Figure 1B). Thus, the method of rocket cultivation had a significant effect on the growth potential of *L. monocytogenes*. In addition, an overall significant difference of *L. monocytogenes* populations on open field rocket (Buzz) was identified between batches (*p* = 0.007). While all three growth potentials from polytunnel rocket (Buzz) were highly similar, there was still an overall significant batch effect detected on *L. monocytogenes* growth (*p* = 0.021). Differences in minimum and maximum daily temperature were also identified between the three batches of both open field and polytunnel rocket (Buzz, ANOVA *p* = 0.006, 0.018, 0.006, and <0.001, respectively Appendix A). Thus, seasonality of batches grown during different time periods played an important role for *L. monocytogenes* populations of open field and polytunnel rocket. Average daily minimum temperatures, maximum temperatures, sunshine exposure, and rain were highest for the Buzz variety open field batch, with the highest growth potential of *L. monocytogenes*. Likewise, the opposite was the case for the batch with the lowest *L. monocytogenes* growth potential (Appendix A). An identical pattern was observed for the Buzz variety cultivated in polytunnel where temperatures followed the growth potentials of *L. monocytogenes*. Additionally, on rocket (Buzz, polytunnel and open fields), a significant overall effect of time on *L. monocytogenes* counts was revealed. Significant differences were identified between all data points from polytunnel samples, in open fields these significant differences were not observed for day 2 to 5 (*p* = 0.071). The second variety of polytunnel rocket Esmee (Figure 2) demonstrated a growth potential of 1.23 log_10_ cfu g^−1^ which was between the average growth potential of polytunnel and open field rocket Buzz. In addition, rocket Esmee demonstrated a similar slow but steady increasing trend of *L. monocytogenes* populations such as on Buzz. 

Polytunnel spinach F1 Trumpet demonstrated growth potentials of 1.45, 1.27, and 1.40 log_10_ cfu g^−1^ (Figure 1C). In comparison, open field spinach F1 Trumpet demonstrated significantly higher growth potentials (average 161% increase) of 2.39, 2.59, and 1.65 log_10_ cfu g^−1^ (*p* = 0.045; Figure 1D). Thus, method of cultivation influenced the growth of *L. monocytogenes* on spinach produce. Polytunnel spinach F1 Trumpet demonstrated growth potentials similar to polytunnel rocket (Buzz). Additionally, between the three independent growth potential batches of polytunnel spinach (F1 Trumpet), dissimilar *L. monocytogenes* growth patterns were displayed (*p* = 0.005), confirming a seasonal effect. This between batch effect was also revealed for open field spinach F1 Trumpet (*p* = < 0.001). As observed with rocket, significant differences were identified between the three batches of both open field and polytunnel spinach (F1 Trumpet) for minimum and maximum daily temperatures (ANOVA *p* < 0.001, <0.001, 0.006 and <0.001, respectively; Appendix A). Thus, seasonality of batches played an important factor in determining populations of *L. monocytogenes* on spinach. Like polytunnel and open field rocket Buzz, highest average daily minimum and maximum temperatures during growth periods was associated with highest growth potentials of *L. monocytogenes* on open field spinach, F1 Trumpet variety. Again, the opposite was the case for the batch with the lowest temperatures that showed the lowest growth potential of *L. monocytogenes* (Appendix A). However, although a seasonality effect occurred, no consistent trend regarding minimum and maximum temperature was found for polytunnel spinach F1 Trumpet variety. As found for rocket Buzz, significant time effects on *L. monocytogenes* growth were identified for polytunnel and open-field spinach F1 Trumpet. Significant differences were identified between all data points except for days 5 to 7 and days 7 to 9 (*p* = 0.166, 0.102, respectively) for polytunnel cultivations, while for open field, significant differences were identified between all data points except from days 5 to 7 (*p* = 0.076). The growth potential demonstrated by the second spinach variety F1 Cello in the polytunnel was 1.84 log_10_ cfu g^−1^ and higher when compared to polytunnel spinach F1 Trumpet. 

Kale cultivated in the polytunnel displayed the highest *L. monocytogenes* growth potential of 2.56 log_10_ cfu g^−1^ (Figure 2) out of all polytunnel (rocket and spinach) produce. However, it led to *L. monocytogenes* growth potentials that were similar to open field spinach F1 Trumpet. Repeated measures ANOVA identified significant differences between all data points; thus, continued growth of *L. monocytogenes* was present throughout this experiment. *L. monocytogenes* populations of spinach F1 Cello began to plateau with no significant differences from days 5 to 7, 7 to 9, and 5 to 9, while rocket Esmee had only no significant differences in *L. monocytogenes* populations from days 5 to 7. Moreover, the plant species, i.e., kale Nero di Toscana, spinach F1 Cello and rocket Esmee (Figure 2), significantly influenced *L. monocytogenes* growth, as revealed by Tukey’s pairwise comparisons (all *p* < 0.001). Similarly, plant species influenced *L. monocytogenes* growth on open field produce as spinach F1 Trumpet and rocket Buzz growth potentials were significantly different (*p* = 0.021). In contrast, plant species did not influence growth potentials of *L. monocytogenes* on polytunnel spinach F1 Trumpet and rocket Buzz (*p* = 0.615). 

### 3.3. L. monocytogenes Growth Curve Parameters from Growth Potential Experiments (Three-Strain L. monocytogenes Mixed Cultures)

The majority of growth curves when fitted to linear models (Appendix A) displayed high R^2^ values. Polytunnel spinach F1 Trumpet displayed the highest variability in the R^2^ values (0.570–0.928). For the remaining growth curves, R^2^ values were only twice just below 0.9 as seen for open field spinach F1 Trumpet (0.871–0.951) and polytunnel rocket Buzz (0.895–0.984). Open field rocket Buzz (0.917–0.957), polytunnel spinach F1 Cello (0.905), polytunnel rocket Esmee (0.975), and polytunnel kale Nero di Toscana (0.988) displayed the highest levels of fit. 

Eight of the fifteen growth curves from the growth potential experiments fitted best to linear models, which described only one growth phase (Appendix A) of which five came from rocket and only three came from spinach. This was followed by five of the fifteen fitting more closely to Baranyi and Roberts (no lag), which described a growth and a stationary phase (three for spinach, one for rocket, and one for kale). One of the fifteen were associated most closely with a biphasic model (no lag), which described growth and stationary phases in two straight lines (rocket). The final one of the fifteen fitted more closely to the Baranyi and Roberts (no asymptote) model, which consisted of only a lag and a growth phase (spinach). 

Based on the most appropriate models (best fitting), the maximum growth rates (µmax, Ln cfu g^−1^ h^−1^) ranged from polytunnel spinach F1 Trumpet (0.0123–0.0520) to open field spinach F1 Trumpet (0.0164–0.0426), polytunnel rocket Buzz (0.0128–0.0202), open field rocket Buzz (0.0095–0.0140), polytunnel spinach F1 Cello (0.0190), polytunnel rocket Esmee (0.0136), and polytunnel kale Nero di Toscana (0.0276) (Appendix A). 

### 3.4. Parameters from Growth Curves of L. monocytogenes on Experimental Farm Produce Inoculated with Single L. monocytogenes Cultures

*L. monocytogenes* counts of separately cultivated strains 959 and 1382 revealed similar growth curves on polytunnel rocket Buzz and spinach F1 Trumpet. At the 216-hour data point (day 9) of the single strains 959 and 1382, the experiments on rocket and spinach *L. monocytogenes* increases (Figure 3) were very similar to the three-strain day 9 growth potentials, as demonstrated on the same produce (Figure 1A,C). 

For *L. monocytogenes* strains 959 and 1382, the growth curves were fitted to the linear, Baranyi and Roberts (no lag), and biphasic (no lag) models. The best fitting model for both strains on spinach was the biphasic (no lag) model, which displayed the highest R^2^ and lowest RMSE (0.950 and 0.106, and 0.946 and 0.114, respectively), with the linear model showing the worst fit (Table 1). Thus, the *L. monocytogenes* growth on spinach was characterised by a stationary phase towards the end of the incubation experiment. In contrast, for rocket, the best fitting model (highest R^2^ and lowest RMSE) for both strains of *L. monocytogenes* was the linear model (0.972 and 0.094 for strain 959; 0.950 and 0.117 for strain 1382). While for strain 1382, worse fitting models could be fitted on rocket, for strain 959 on rocket, ComBase was not able to fit any other model than the linear one (Appendix A).

Based on the best fitting models, of all maximum growth rates tested on polytunnel products, *L. monocytogenes* strain 959 on spinach displayed the largest overall, i.e., 0.0197 Ln cfu g^−1^ h^−1^. This was followed by *L. monocytogenes* strain 959 on rocket (0.0160 Ln cfu g^−1^ h^−1^); *L. monocytogenes* strain 1382 on spinach (0.0157 Ln cfu g^−1^ h^−1^); and lastly, *L. monocytogenes* strain 1382 on rocket (0.0147 Ln cfu g^−1^ h^−1^) (Appendix A).

### 3.5. Comparison of Experimental Farm Leafy Vegetables’ Total Bacteria Counts (TBCs) and Subsequent Influence on Growth of L. monocytogenes

Plant species greatly affected the total bacterial counts (TBCs) of all tested vegetables. Spinach displayed highest initial TBCs of 7 log_10_ cfu g^−1^ (Figure 4C,D and Figure 5), followed by rocket, which had TBCs between 5 and 6 log_10_ cfu g^−1^ (Figure 4A,B and Figure 5), and kale’s TBCs started between 2 and 3 log_10_ cfu g^−1^ (Figure 5). Likewise, significant differences in TBCs were found when kale (Nero di Toscana) was compared to spinach F1 Cello and rocket Esmee (Figure 2; Tukey’s pairwise comparisons, all *p* < 0.001). Significant differences were identified between the TBCs of the batches of *L. monocytogenes*-inoculated produce. The largest difference was demonstrated by open field rocket Buzz, followed by open field spinach F1 Trumpet; then, polytunnel rocket Buzz; and finally, polytunnel spinach F1 Trumpet (all *p* < 0.001). Furthermore, for TBCs of open field spinach F1 Trumpet and polytunnel rocket Buzz, significant differences were identified between all data points (time effect) other than days 5 to 7 (*p* = 1.000 and 0.071, respectively). A similar time effect was observed for polytunnel spinach F1 Trumpet with the exceptions of days 5 to 7 and days 7 to 9 (*p* = 0.959 and 0.119, respectively). In contrast, significant differences were identified between all data points on open field rocket (Buzz). TBCs of polytunnel rocket Esmee (Figure 5) started at similar counts (day 0) as batch 3 of polytunnel and batch 1 of open field rocket Buzz (Figure 4). In contrast, spinach polytunnel F1 Cello (Figure 5) displayed different TBCs to spinach F1 Trumpet (Figure 4). In comparison to spinach F1 Trumpet and rocket Buzz, all three batches of inoculated polytunnel produce in Figure 5, had more non-significant differences in TBCs. In particular, all three batches shared no significant increases in TBCs from days 0 to 2 and 7 to 9. 

The effect of *L. monocytogenes* inoculation on TBCs of all produce tested in this study appeared to be negligible. TBCs with and without *L. monocytogenes* were extremely similar for all comparisons. Nevertheless, low TBCs of kale were associated with higher levels of *L. monocytogenes* growth (Figure 2). In contrast, lower TBCs of batch three open field spinach F1 Trumpet (Figure 4D) were associated with considerably lower levels of *L. monocytogenes* growth than found among the other two batches (1.65 vs. 2.59 and 2.39 log_10_ cfu g^−1^, Figure 1D). 

Growth curve fitting for TBC was conducted in the same way as for the *L. monocytogenes* three-strain mix experiments. While linear models had a fit of between 0.41 and 0.94, the use of either linear, biphasic or Baranyi and Roberts models improved the fitting range of no less than 0.76 and up to 0.99. The best fits for six out of seven spinach incubation experiments were all based on a stationary phase towards the end of the experiment (Baranyi and Roberts (complete or no lag) or biphasic), with the remaining one best fit to Baranyi and Roberts (no asymptote) without a stationary phase. However, for rocket, this was only the case for one (Esmee variety) out of seven incubation experiments with the remaining six (all Buzz variety) fitted best applying to linear models and Baranyi and Roberts (no asymptote) models without a stationary phase. Thus, the results from spinach indicate that by majority total bacterial density reached a plateau. In contrast, total bacterial density on rocket showed by majority maximum and no signs of inhibition of growth after 9 days of incubation (Appendix A).

## 4. Discussion

The current study sought to describe how vegetable species and variety, method, and seasonality of cultivation affects the growth of *L. monocytogenes* on leafy vegetables. Furthermore, this study recorded the growth of the indigenous vegetable phyllosphere bacteria alongside the growth of *L. monocytogenes*. All tested cultivation variables appeared to be at least co-factors for *L. monocytogenes* growth and may contribute to the apparent contradictions identified among preceding reports. 

As confirmed in the present study, the growth of *L. monocytogenes* is dependent on plant species. These different rates of growth or even the lack of growth have been well reported in the literature, where experimental and pre-harvest conditions varied considerably between the reported studies putatively affecting the outcomes [18,19,21,22,23,37,38]. Similarly, albeit at high inoculation levels, growth and survival of a gfp marked *E. coli O157:H7* strain appeared to be highly dependent on the plant species, i.e., baby leaf spinach, rocket, and Swiss chard [39]. However, this difference was not observed at the cultivar level. In a contrasting study, *E. coli O157:H7* persistence at 6.5 log_10_ cfu per spinach leaf was influenced by the organic spinach cultivar, i.e., Emilia, Waitiki, Lazio, and Space. This was attributed to differences in leaf surface roughness and stomata density, where higher degrees of growth were associated with greater surface roughness [40]. 

Differences in leaf surface of baby spinach (F1 Trumpet vs. Cello) and rocket (Buzz vs. Esmee) varieties may have also contributed to the differences in growth *of L. monocytogenes* in the current study while being under the same cultivation conditions. In addition, spinach cultivars have been shown to be an influencing factor for the population size and community structure (diversity) of the phyllosphere [41]. Spinach variety savoy was found to have a larger leaf surface area containing more stomata and glandular trichomes. These structures contained increased levels of bacterial aggregates and were associated with larger microbial populations and increased richness of phyllosphere, in comparison to flatter spinach leaf varieties. Lopez-Velasco and colleagues (2011) speculated that the same factors that influence the phyllosphere diversity could also co-determine growth of the likes of *Listeria* and *E. coli*. Other surface characteristics including leaf hydro-phobicity or hydro-philicity may also influence the attachment of *L. monocytogenes*. Growth on hydrophobic leaf surfaces such as spinach and kale (109.57° and 109.84°, respectively) result in lower initial microbiota concentrations compared to more hydrophilic leaf surfaces including romaine lettuce (57.60°) [42]. However, the present authors have found no differences in bacterial counts on spinach and lettuce in a previous study [21]. In the current study, all produce including kale, rocket, and spinach retained the initial target inoculum of 100 cfu g^−1^, which did not indicate a lack of attachment. The ability of *L. monocytogenes* to attach to surfaces such as vegetables can be linked to various properties that includes motility as well as internalin A and genes such as luxS [43,44]. It has been reported that the attachment of pathogenic bacteria including *L. monocytogenes* can be strain dependent [45]. Indeed, the current study was performed with a three-strain *L. monocytogenes* mix to account for variations in growth among strains. 

A relative increase between 1.2 and 1.5 log_10_ cfu g^−1^ in *L. monocytogenes* populations from day 9 to 10 was detected on kale during 7 °C incubation [22]. Moreover, single *L. monocytogenes* inoculated the maximum growth rate experiments on rocket in this study identified increases from day 9 to day 10. Therefore, extending the incubation period in the current study could potentially lead to the recording of even higher growth potentials, which was limited to nine days for the three-strain inoculation. Brassicaceae species such as kale contain secondary metabolites known as glucosinolates, which produce antimicrobial activity towards plant pathogens [46]. Rocket was also revealed to contain 238, 145, and 397 μmol per 100 g^−1^ dry weight of glucoraphanin, gluconasturtiin, and 4-methoxyglucobrassicin, respectively [47]. Allyl isothiocyanate in free form or as glucosinolates, which is found in cruciferous plants (kale and rocket), has demonstrated inhibitory effects against *L. monocytogenes* on food in conditions ≤21 °C at a neutral pH [48]. However, no clear inhibition of *L. monocytogenes* was observed on those products in the present study. One may speculate that this could be another variety specific factor, determining the growth of *L. monocytogenes*. 

Maximum growth rates of both *L. monocytogenes* strains on rocket and spinach were similar in this study. However, on average, these growth rates appear to be slightly lower than those published on leafy vegetables at similar storage conditions. Maximum growth rates of *L. monocytogenes* on spinach at 5 and 8 °C were 0.024 and 0.037 log_10_ cfu g^−1^ h^−1^, respectively [30]. Be that as it may, Omac and colleagues’ (2018) growth curves are based on a 400-hour period compared to 240 h in the present study; therefore, increases in growth after this period (240 h) may explain their higher slope (maximum growth rate). On an iceberg and crisp lettuce mix, maximum growth rates of *L. monocytogenes* were on average 0.02 log_10_ cfu g^−1^ h^−1^ at 7 and 10 °C [32]. Additionally, those authors identified no significant differences between the maximum growth rates of three different *L. monocytogenes* strains, which was reflected in the current study. 

In the present study, polytunnel and open field cultivation methods led to significantly different growth potentials (*p* < 0.05) on rocket (Buzz variety) and spinach (F1 Trumpet variety). Dissimilarly, Gutiérrez-Rodríguez and colleagues found no difference in *E. coli* and *E. coli O157:H7* populations between spinach produce grown in a greenhouse (hydroponic) and open field setting [49]. Protein as a source of nitrogen, sugars, and other inorganic nutrients are localised within cellular tissues of leafy vegetables [50,51]. Therefore, when *L. monocytogenes* is inoculated onto an undamaged intact leaf surface, it cannot utilise these nutrients for growth. However, weather conditions of high humidity (temperate climates) or rain due to presence of liquid water on the exterior of the leaf cuticle causes leaching of nutrients from the leaf endosphere to the leaf surface (phyllosphere) [51,52,53], thus, providing nutrients such as extracellular polysaccharides which *L. monocytogenes* utilises for colonisation [54]. Therefore, this could be a contributing factor why cultivation method caused a difference in this study, i.e., open field spinach had higher growth of *L. monocytogenes*. Additionally, leaching increases with leaf age due to the greater wettability of older leaves [51]. This agrees with the current study as more nutrients are present on older leaves causing higher *L. monocytogenes* growth, which was observed for open field spinach. Open field spinach had on average longer growing periods compared to polytunnel spinach due to lower air temperatures. 

However, unexpectedly, the less humid environment within the polytunnel setting had higher *L. monocytogenes* growth for rocket even though rocket has a similar nutrient (protein, carbohydrate, and fat) content when compared to spinach (3.6, traces, and 0.4 g 100 g^−1^ vs. 2.8, 1.6, and 0.8 g 100 g^−1^, respectively) [55]. However, this could be owing to leaf physiology, i.e., the greater surface area of spinach leaves compared to rocket. Indeed, wider leaves with larger surface areas are associated with higher presence of stomata and, thus, increased rates of transpiration causing leaching of nutrients from the leaf endosphere [56] for the potential benefit of *L. monocytogenes*.

A significant effect between batches was identified for all cultivation methods for both rocket (Buzz) and spinach (F1 Trumpet), thus implying that a seasonality effect was omnipresent in the current study. This can be supported by differences in average maximum and minimum temperatures, and sunlight along with precipitation between batches in the current study, which is typical for moderate climates with a strong Atlantic influence. A recent study has revealed that seasonality impacts the phyllosphere bacterial community of the spinach leaf [57]. In that study, 709 operational taxonomic units (OTUs) were affected by seasonality, i.e., increasing average air temperatures between experiments: 15.9 ˚C, 17.9 ˚C, and 20.2 ˚C. Of these 709 OTUs, 255 were negatively affected by season, whereas 454 were significantly enriched. The same authors suggested that drought can lead to decreased diversity of the phyllosphere, thus resulting in proliferation of pathogenic microorganisms and increased persistence of an inoculated *E. coli O157:H7* strain. Furthermore, subjecting spinach leaves to lower air and soil temperatures and decreased precipitation have been linked to smaller microbial population counts and reduced bacterial community richness [41]. These studies could explain the differences in *L. monocytogenes* growth and TBCs due to the seasonality as well as the cultivation method in the current study. Subjecting red-pigmented baby leaf lettuce during growth period to differing levels of photosynthetically active radiation significantly influenced the bacterial community structure but not the total bacterial abundance [58]. A separate study on baby leaf spinach suggested that wind speed, solar radiation, and relative humidity are influential factors of total microbial population, Enterobacteriaceae, and *Pseudomonas* counts during the growth of produce [59]. Therefore, the bacterial community structure of the phyllosphere of open field and polytunnel produce in this study are likely to differ significantly due to protection from photosynthetically active radiation, wind, and precipitation within the polytunnel setting. Moreover, the intensity of these weather parameters, i.e., sunlight, air temperature, humidity, and wind speed all impact the amount as well as the opening and closing of stomata on leaves, which influences leaching of nutrients (speed of transpiration rate), which are ultimately utilised by *L. monocytogenes* for growth [56]. Indeed, in the present study, faster growing rocket due to increased temperatures and sunlight were associated with higher *L. monocytogenes* growth potentials. As previously discussed, rocket leaves contain on average fewer stomata and grandular trichomes than spinach, thus leaching is of lesser importance to understanding the growth of *L. monocytogenes* on rocket leaves. Therefore, the increased growth of *L. monocytogenes* on rocket could be explained by climatic conditions enhancing the growth of phyllosphere bacterial community members, which in-turn cause a proliferation of *L. monocytogenes.*

In a previous study, lettuce leaves were inoculated with 3 log_10_ cfu g^−1^ of *L. monocytogenes*, while the leaves had a TBC of 5.61 log_10_ cfu g^−1^. At this 1:100 ratio, the TBC did not affect the growth rate of *L. monocytogenes* at 5, 10, 15, 20, and 25 ˚C temperature profiles [50]. However, when a 1:1 (*L. monocytogenes*-to-TBC) ratio was established with an acidic electrolysed water leaf treatment, the maximum population densities of *L. monocytogenes* on lettuce was higher for all temperatures when compared to the 1:100 ratio. Altering starting microbial populations with chlorine dipping to 4.7, 5.5, and 6.5 log_10_ cfu g^−1^ of conventional and 5.3, 6.1, and 6.7 log_10_ cfu g^−1^ of organic romaine lettuce did not impact *L. monocytogenes* populations throughout storage at 10 °C when inoculated at 10^5^ CFU g^−1^ [60]. Moreover, at 5 °C, an initial concentration of 2.08 log_10_ cfu g^−1^ led to an estimated maximum *L. monocytogenes* population density of 4.15 log_10_ cfu g^−1^, and at 10 °C, 2.55 log_10_ cfu g^−1^ led to 5.61 log_10_ cfu g^−1^. A later study from the same authors showed that total aerobic mesophilic and psychrotrophic bacteria of fresh spinach leaves limits the growth of *L. monocytogenes* at temperatures ranging from 3 to 8 °C [30]. Indeed, in the current study, competition between TBCs and *L. monocytogenes* populations occurred, particularly for spinach. However, this effect was not as consistent for rocket. Moreover, higher TBCs products spinach and rocket were associated with lower *L. monocytogenes* growth than the lower TBC-associated produce kale. It can be speculated that the effect of the TBCs is either dependent on the plant species and/or the microbial community structure in the phyllosphere. Due to these observations, it is important to establish the identities of phyllosphere bacterial communities of these leafy vegetables in the future. 

Phyllosphere bacterial communities and individual strains of microorganisms have exhibited resistance towards pathogens through the production of antimicrobial compounds referred to as the Jameson effect, i.e., inter-microbial competition [61]. In fact, *L. monocytogenes* growth is more susceptible to becoming restricted due to the presence of inhibiting microorganisms associated with minimally processed leafy vegetables when compared to *Salmonella* [18]. The iceberg lettuce community structure, contaminated with *L. monocytogenes* and treated with 5 mg kg^−1^ of nisin, constantly changed over 7 days (air, 4 °C), with Pseudomonadaceae having greatest relative abundance overall, followed by Enterobacteriaceae on day 0 and Streptococcaceae and Lactobacillaceae on days 2 and 5 [26]. However, the same authors failed to establish a culture of nisin producing *Lactococcus lactis* on the same lettuce to restrict growth of *L. monocytogenes* directly. Bacterial isolates from lettuce have been shown to influence the growth of *Listeria* in co-cultures [62]. Less is known about the in-situ influence of the other leafy vegetable phyllosphere bacterial communities when it comes to the growth of *L. monocytogenes*. However, the in-situ influence of rocket’s phyllosphere against *L. monocytogenes* appears to be promising as a bacteriocinogenic strain of *Lactococcus lactis subsp. Lactis* has been isolated from rocket salad (Brazil), which produced lantibiotic, an antimicrobial nisin variant [63]. Moreover, *Lactiplantibacillus plantarum*, another lactic acid bacteria, was isolated from bagged rocket [64]. Those authors identified that the strain harbours genes that encode the production of Coagulin A and Pediocin ACH. Coagulin A immobilises invading bacteria, inhibiting their spread, while Pediocin ACH is an active peptide that displays highly specific inhibition of *L. monocytogenes* [65,66]. It has been reported that such lactic acid bacteria are detected more frequently on RTE leafy vegetables throughout spring and summer as opposed to autumn and winter [67]. 

In addition, the *Lactobacillus plantarum* strain SK1 exhibits anti-listerial activity due to the production of lactic acid, which could potentially act as biological control agent on leafy vegetables [68]. Indeed, there were significant differences in pH over course of the current experiments, albeit no singular trend was identified, and pH values remained at optimal conditions for *L. monocytogenes* growth [14]. pH values of raw leafy vegetables can be reduced due to lactic acid fermentation by presence of lactic acid bacteria in bacterial communities under favourable conditions of anaerobiosis, water activity, salt concentration, and temperature [69]. However, the current study employed ambient atmospheric conditions that did not promote lactic acid fermentation. 

## 5. Conclusions

The present study has demonstrated that the vegetable species and cultivar, the method of cultivation in the form of open field or polytunnel use, and seasonality of cultivation all affected growth of *L. monocytogenes*. While in the present study, all growth potentials were above 0.50 log_10_ cfu g^−1^, this may not be the case for other food products. Therefore, the cultivation conditions or variety selection has the potential to determine whether a food product is considered to support growth. Consequently, there is the risk that food products that are identified to not support growth under one set of environmental cultivation conditions may support growth under different cultivation conditions. This could pose as additional risks in ready-to-eat food productions for both the producers and the consumers. Policy-makers should therefore consider further revising the EU guidelines for conducting growth potential studies to include seasonality effects on certain food products such as leafy vegetables.

When considering the cultivation methods chosen in the present study, open fields for rocket (Buzz) and polytunnel for spinach (F1 Trumpet) can be recommended for growers as they had consistently significantly lower growth potentials. For recommendations on variety, polytunnel rocket (Esmee) and spinach (F1 Trumpet) were associated with lower growth potentials. However, further analyses of polytunnel spinach (F1 Cello) and rocket (Esmee) would need to be conducted to substantiate such recommendations.

Future studies should perform in depth phyllosphere community analyses by employing methods such as next-generation sequencing to determine differences in the development of the phyllosphere microbiome. Changes in bacterial community structures due to vegetable species and variety, cultivation method (i.e., polytunnel versus open field), and seasonality of cultivation may be important co-factors for differing levels of *L. monocytogenes* growth, as observed in this study.

## Figures and Tables

**Figure 1 foods-11-03056-f001:**
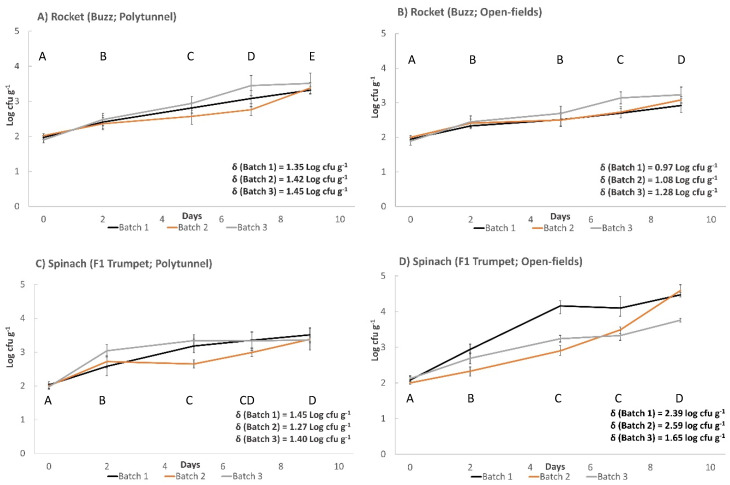
Growth of *L. monocytogenes* (100 cfu g^−1^ initial inoculum) on experimental farm produce. (**A**) Polytunnel rocket (Buzz). (**B**) Open field rocket (Buzz). (**C**) Polytunnel spinach (F1 Trumpet). (**D**) Open field spinach (F1 Trumpet). Incubated at 7 °C for (6 days) followed by 12 °C (3 days). (±) error bars indicate standard deviation; δ represents growth potential value for each batch; letters A–E indicate significant differences over time.

**Figure 2 foods-11-03056-f002:**
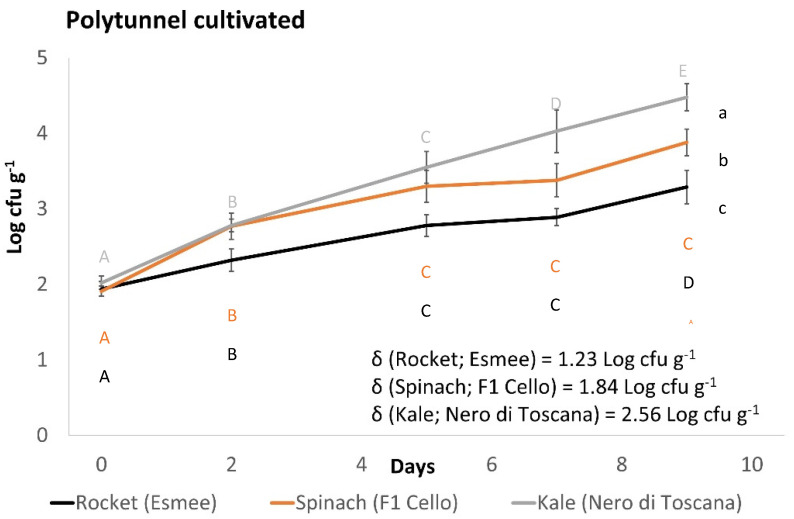
Growth of *L. monocytogenes* (100 cfu g ^−1^ initial inoculum) on polytunnel produce: rocket (Esmee) (black line), spinach (F1 Cello) (brown line) and Kale (Nero di Toscana) (grey line). Incubated at 7 °C for (6 days) followed by 12 °C (3 days). (±) error bars indicate standard deviation; δ represents growth potential value for each batch; letters a–c indicate significant differences in *L. monocytogenes* growth; letters A–E indicate significant differences over time for rocket (black), spinach (brown), and kale (grey).

**Figure 3 foods-11-03056-f003:**
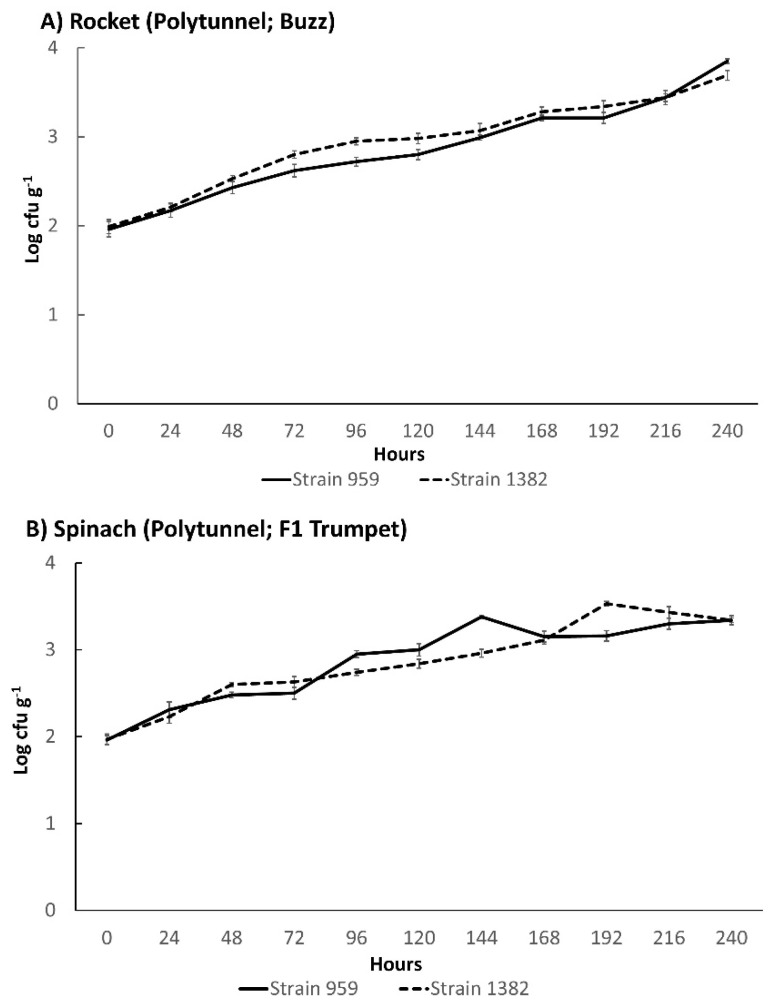
Growth curves of single *L. monocytogenes* strains 959 (Solid line) and 1382 (Dashed line) on experimental farm produce: (**A**) polytunnel rocket (Buzz) and (**B**) polytunnel spinach (F1 Trumpet). Incubated at 8 °C for 10 days. (±) error bars indicate standard deviation.

**Figure 4 foods-11-03056-f004:**
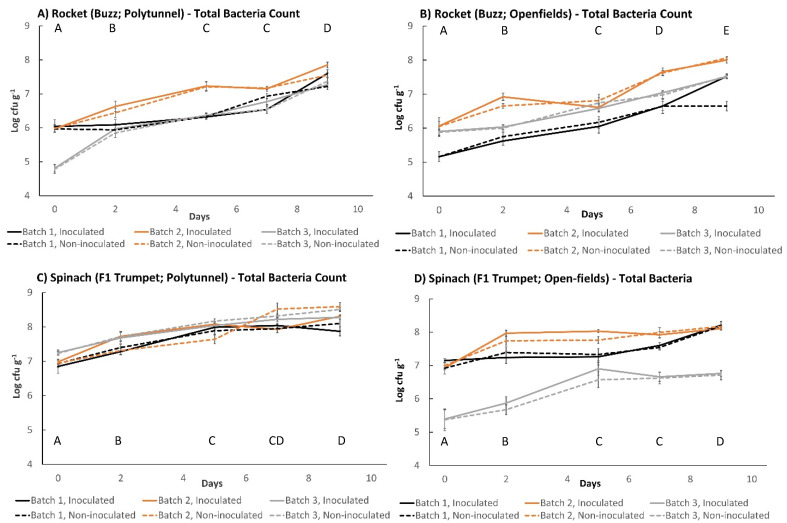
TBCs of experimental farm produce: (**A**) polytunnel rocket (Buzz), (**B**) open field rocket (Buzz), (**C**) polytunnel spinach (F1 Trumpet), and (**D**) open field spinach (F1 Trumpet). Incubated at 7 °C for (6 days) followed by 12 °C (3 days). (±) error bars indicate standard deviation. Solid lines represent produce which has been initially inoculated with 100 cfu g^−1^ of *L. monocytogenes*; dashed lines represent test units which have not been initially inoculated; letters A–E indicate significant difference over time in batches inoculated with *L. monocytogenes*.

**Figure 5 foods-11-03056-f005:**
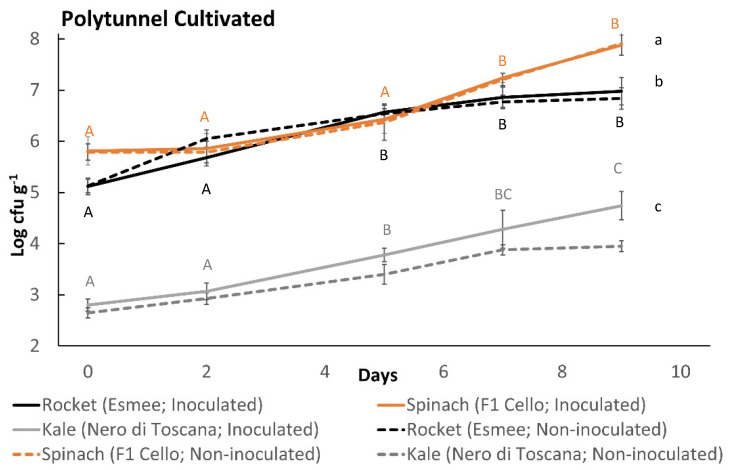
TBCs of experimental farm polytunnel produce: rocket (Esmee, black line), spinach (F1 Cello, brown line), and kale (Nero di Toscana, grey line). Incubated at 7 °C for (6 days) followed by 12 °C (3 days). (±) error bars indicate standard deviation. Solid lines represent produce which has been initially inoculated with 100 cfu g^−1^ of *L. monocytogenes*, dashed lines represent test units that have not been initially inoculated. Letters a–c indicate significant differences in *L. monocytogenes* growth on inoculated produce; letters A–C indicate significant difference over time in batches inoculated with *L. monocytogenes* for rocket (black), spinach (brown), and kale (grey).

**Table 1 foods-11-03056-t001:** Maximum growth rates of *L. monocytogenes* strains 959 and 1382 on experimental farm Spinach (F1 Trumpet) and Rocket (Buzz).

Product	Strain	Model	R^2^	RMSE	Maximum Growth Rate (μmax) Ln cfu g^−1^ h^−1^
Spinach	959	Linear	0.836	0.193	0.0127
Spinach	959	Baranyi and Roberts (no lag)	0.942	0.115	0.0211
Spinach	959	Biphasic model (no lag)	0.950	0.106	0.0197
Spinach	1382	Linear	0.916	0.142	0.0136
Spinach	1382	Baranyi and Roberts (no lag)	0.934	0.125	0.0158
Spinach	1382	Biphasic (no lag)	0.946	0.114	0.0157
Rocket	959	Linear	0.972	0.0935	0.0160
Rocket	1382	Linear	0.950	0.117	0.0147
Rocket	1382	Baranyi and Roberts (no lag)	0.949	0.118	0.0162
Rocket	1382	Biphasic (no lag)	0.943	0.124	0.0148

## Data Availability

Data is contained within the article or Appendix A.

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
