# Peer review of "Cultivation Conditions of Spinach and Rocket Influence Epiphytic Growth of Listeria monocytogenes"

_foods, 2022, doi:10.3390/foods11193056_

Round 1

Reviewer 1 Report

Dear editor and author, 

Major comments

There are some species bacteria belonging to the Listeria genus that cause food poisoning such as Listeria innocua and why this species has not been evaluated?

Minor comments

1-The full name of the bacteria must be written in the title of the manuscript.

2-The number of bacteria must be added to units (CFU/ g).  see line 13.

3-The introduction of the manuscript needs to add the reference in line 103, I suggest ( Ross, T., & McMeekin, T. A. (2003). Modeling microbial growth within food safety risk assessments. Risk Analysis: An International Journal23(1), 179-197.)

4-The name of the bacteria is not written in italics in some of the sub-headings, see line 133 and line 201.

5- Total Bacteria Counts (TBC) need to add reference, I suggest (Detection of Listeria monocytogenes bacteria in four types of milk using PCR. Pakistan Journal of Nutrition11(12), 1158.)
6-The Y-axis title should be modified in Figures 1,2 , 3, 4 and 5 to Log CFU.g-1 .
7- Supplementary tables contain some unknown symbols, the symbols are usually known below the tables.

8-The conclusions must be rewritten, Conclusions contain several results that must be removed.

Author Response

Major comments

There are some species bacteria belonging to the Listeria genus that cause food poisoning such as Listeria innocua and why this species has not been evaluated?

Listeria innocua is a species that is generally considered as non-pathogenic. Only a handful infections are reported by L. innocua of immunocompromised individuals. Indeed, numerous studies have used L. innocua as a non-pathogenic surrogate strain for L. monocytogenes that is responsible for Listeriosis in humans. We therefore have concentrated our work on L. monocytogenes.

Minor comments

1-The full name of the bacteria must be written in the title of the manuscript.

This has been corrected now (L3 of final version).

2-The number of bacteria must be added to units (CFU/ g).  see line 13

This has now been changed throughout the abstract as suggested (L12-18 of final version).

3-The introduction of the manuscript needs to add the reference in line 103, I suggest ( Ross, T., & McMeekin, T. A. (2003). Modeling microbial growth within food safety risk assessments. Risk Analysis: An International Journal23(1), 179-197.)

Thank you for the suggestion, this has now been added (L95 of final version).

4-The name of the bacteria is not written in italics in some of the sub-headings, see line 133 and line 201.

Thank you for highlighting this. Some journals prefer to have genus and species names not italized when the whole title is in italics. This seems to be not the case here, we have made the changes now throughout the manuscript (L125, 194, 304, 390, 412. 446 of final version).

5- Total Bacteria Counts (TBC) need to add reference, I suggest (Detection of Listeria monocytogenes bacteria in four types of milk using PCR. Pakistan Journal of Nutrition11(12), 1158.)

Thank you for the suggestion. However, we think that the following reference may be better suited:

Johnston, L. M., et al. (2005). "A field study of the microbiological quality of fresh produce." Journal of Food Protection 68(9): 1840-1847. (L248 of final version)

6-The Y-axis title should be modified in Figures 1,2 , 3, 4 and 5 to Log CFU.g-1 .

Thank you for highlighting this. However, we would like to keep the format as in our 2020 paper. We have now improved the figure resolution so that the labelling can be properly assessed (all figures).

7- Supplementary tables contain some unknown symbols, the symbols are usually known below the tables.

We assume that growth potential and max growth rate symbols are meant here. We have added now more details to the table descriptions in Appendix A.

8-The conclusions must be rewritten, Conclusions contain several results that must be removed.

We have now rewritten section 5 as suggested (L689-712 of final version).

Reviewer 2 Report

The prevalence of L. monocytogenes was confirmed on a variety of leafy vegetable samples, so it is essential to study the effect of vegetable culture conditions on L. monocytogenes. And this study hypothesized that different culture conditions would lead to conflicting growth reports on rocket and spinach. Therefore, in this study, cultivation factors, i.e., vegetable species and variety, open field versus poly-tunnel cultivation, and seasonality on the growth of L. monocytogenes on leafy vegetables were tested. Recommendations are made for leafy vegetable producers and EU guidelines for conducting growth potential studies to reduce L. monocytogenes growth levels on spinach and rocket products. In general, there are some minor issues in this manuscript that need to be addressed as follows.

1. In the manuscript, lines 188-200 and 285-306 describe that the pH and water activity of vegetable products have no effect on L. monocytogenes, but the reviewers suggest that the authors should compare the results with other literature to justify the need Conduct research on this. In lines 300-306, water activity and pH changes for all products are shown within the range of L. monocytogenes survival and growth conditions, as described in lines 61-63. Therefore, from the range of L. monocytogenes survival and growth conditions, it can be clearly inferred that the water activity and pH of leafy vegetables do not affect its growth. So the reviewer considers this part to be unnecessary in the manuscript as well.

2. The description of the results should be clearer, but not verbose. The reviewers recommend that the authors further analyze the relationship between the different experimental results, rather than just listing all the resulting data. As shown in 3.2 "Growth of Listeria monocytogenes on experimental produce" and 3.5 "Total bacterial count (TBC) of experimental produce". Furthermore, the digital pixels in this manuscript are not sharp enough.           

Reviewer 3 Report

The authors focused on the influence of cultivation conditions on the epiphytic growth of Listeria monocytogenes in leafy vegetables. This research is interesting and useful to the food community. However, the manuscript needs to be improved in terms of data analysis and result presentation to meet the high expectations of the journal. Here is the list of comments and suggestions:

(1) Please spell out the bacterium name “L. monocytogenes” in the title.

(2) Lines 11-13: Was the difference in bacterial population on various leafy vegetables significant?

(3) Lines 14-16: The difference in bacterial population was at a low number (0.3 log units only), which did not seem to be significant.

(4) Line 25: Reorganization of the Introduction section is recommended. Please delete some parts that are not closely relevant to this study.

(5) Lines 81-83: What are the reasons for these contradicting observations?

(6) Lines 141-142: Why was L. monocytogenes incubated at 7 °C for 7 days to prepare bacterial suspensions? Under this condition, the number of bacterial cells may be low and can not be accurately determined by spectrophotometry.

(7) Lines 147-151: Please indicate the cultivation conditions for maximum growth of L. monocytogenes.

(8) Lines 242-243: Please provide the target gene for the primers.

(9) Lines 256-257: TBC values on other days (i.e., days 2, 5, and 7) should also be presented in order to keep in accordance with the related tests.

(10) Lines 263-264: How many biological and technical replicates were set?

(11) The figures were not clear and should be reproduced. Moreover, the statistical difference needs to be marked in the figures.

(12) Please discuss the results in-depth. For example, which kind of attachment genes in L. monocytogenes may contribute to the difference in attachment ability as indicated in lines 518-521?

(13) The conclusion should be clear and concise.

(14) Lines 675-681: Future research perspectives should be combined with those in lines 657-662. Please rewrite.

Round 2

Reviewer 1 Report

Dear Editor(s), 

The authors made all the necessary changes to improve the manuscript, and now I recommend it for publication in its current form.

Reviewer 3 Report

This manuscript is acceptable in its current form. However, I would suggest the authors provide high-resolution figures for production.